# Tumor Necrosis Factor: What Is in a Name?

**DOI:** 10.3390/cancers14215270

**Published:** 2022-10-27

**Authors:** Xinming Wang, Chunlan Yang, Heinrich Körner, Chaoliang Ge

**Affiliations:** 1Department of Pharmacy, First Affiliated Hospital of Anhui Medical University, Hefei 230022, China; 2Menzies Institute for Medical Research, Liverpool Street, Hobart, TAS 7000, Australia

**Keywords:** TNF, inflammation, cancer

## Abstract

**Simple Summary:**

Cytokines are small molecules that modify the response of the human body to external signals. Tumor Necrosis Factor was found in 1975 and its main activity was assumed to be lethal for cancer cells and to be able to kill tumors, hence the name. Research over the last 35 years has shown that this is not really the case. This review illuminates what caused this misunderstanding and answers the question if there is any connection at all between the cytokine Tumor Necrosis Factor and cancer.

**Abstract:**

Tumor Necrosis Factor was one of the first cytokines described in the literature as a soluble mediator of cytotoxicity to tumors. Over the years, more extensive research that tried to employ Tumor Necrosis Factor in cancer treatments showed nevertheless that it mainly functioned as a proinflammatory cytokine. However, this did not stop the search for the holy grail of cancer research: A cytokine that could act as a one-stop treatment for solid tumors and lymphomas. This review will summarize the long experimental history of Tumor Necrosis Factor that caused the initial observations of a tumor necrotizing cytokine that could serve as a potential cancer treatment and discuss the current state of research into this side of the activities of Tumor Necrosis Factor.

## 1. Introduction

The cytokine Tumor Necrosis Factor is a central immune mediator with a wide variety of functions. It has been shown to be essential in the early, inflammatory aspect of the immune response [1] but also has prominent roles in immune regulation [2], immune defense [3], immune cell differentiation [4], fat metabolism [5], and neurobiology [6]. This review will discuss its name-giving role in cancer biology and defense against malignant cell growth.

The name *Tumor Necrosis Factor* (TNF) can be dated back to experiments published by Carswell, Old et al. in 1975, who showed that in Bacille Calmette–Guerin (BCG)-infected and subsequently LPS-challenged rabbits, a tumor-necrotizing agent existed that was transferable [7]. Later it was demonstrated that this unpurified and unspecified agent that caused the necrosis of methylcholanthrene-(Meth-A)-induced sarcomas in mouse models [7] was also cytotoxic for transformed WEHI-164 cells [8], which allowed for a simple method of detection using cell culture. Interestingly, the idea that a tumor-necrotizing agent existed that was expressed in response to bacterial infection or the inoculation of bacterial components into solid tumors dated back much further. In the early 1890s, William B Coley, a surgeon in New York, began treating cancer patients with a concoction of extracts from different bacterial species [9]. This treatment was based on anecdotes, including William Coley’s personal experiences, that described spontaneous tumor remission after a bacterial infection of seemingly doomed patients. The interesting story of “Coley’s mixed toxin” is not the subject of this review but can be read by his own account in a talk he gave to the Royal Society, where he summarized his journey of discovery [10], and in seminal reviews by Lloyd Old [11] and Frances Balkwill [12].

The treatment was controversial, and reports of its effectiveness as reviewed 40 years later by William Coley’s daughter Helen Coley Nauts were only based on anecdotal evidence. It was, however, still available until 1990 when its production ceased because it missed the approval of regulatory authorities, but it can to this day be obtained as a specialized fever therapy for cancer (Fieberterapie) in Germany.

The contemporary view of TNF is that it is a proinflammatory cytokine. Generally, TNF is expressed in large amounts early in the immune response, predominantly by macrophages [13] but also produced by NK cells [14], B cells [15], and T lymphocytes [16]. The various activities of the cytokine have been investigated in models of inflammation, cancer, autoimmunity, and infection using gene-deficient mice [17,18,19], and more recently, in tissue-specific gene-deficient mouse models unable to express TNF in either macrophages or T cells [20].

A vast spectrum of functions and activities in host defense and the initiation and maintenance of inflammation was demonstrated [21] and “TNF blockade” is commonly used in anti-inflammatory therapies for chronic inflammatory diseases such as rheumatoid arthritis (RA) [22]. Nevertheless, the question if and how this cytokine can modify the chance of developing cancer and how anti-TNF therapy could interfere with the natural anti-tumor response, depending on the immunological context, is still being discussed in the medical community. This review will present an updated summary that explores the potential role of TNF in cancer and the various underlying mechanisms that could explain an interference.

## 2. A Short History of TNF

After Coley’s original observation that solid tumors treated with bacterial components could develop hemorrhagic necrosis, clinical advances in cancer treatment caused the interest in this line of research to subside. However, various groups persistently tried to identify the factor that caused the necrotizing of tumors. In 1968, a group was successful in isolating a serum component that was cytotoxic for a transformed cell line [23]. A few years later, the name Tumor Necrosis Factor was coined for a serum factor that caused the destruction of tumors in vivo and was also active against tumor cell lines in vitro [7,24]. After the successful cloning of the human [25] and mouse genes [26,27] for TNF and its sister molecule Lymphotoxin α (LTα) [28,29,30,30] and the availability of recombinant proteins, the search for a receptor was soon successful [31,32,33]. In fact, it became clear that two different receptors existed that bound TNF as well as LTα yet displayed functional differences [34,35,36]. Only one receptor termed TNF-Receptor-1 (TNFR-1) (p55) was shown to be important for the protection against bacterial infection and the proinflammatory activity of endotoxins [37,38,39], while the role of the second TNFR-2 p75 remains to some extent unresolved to this day, despite a demonstrated function in cell proliferation [40], regulatory T cell proliferation [2], and general support for p55 [41,42]. The immediate TNF family (TNF and LTa) was extended with the discovery of a new partner of LTα, LTβ [43,44]. In the form of a heterotrimer, LTα^2^ × LTβ bound to a new receptor termed LTβR [45]. Historically, TNF was termed TNF-α to differentiate it from TNFβ (LTα). The discovery of LTβ and the subsequent use of LTα instead of TNFβ made the use of TNF-α redundant. In our review, we will refer to the cytokine as “TNF”.

Gene-deficient mouse models of both receptors, p55 [38,39] and p75 [46,47], of TNF [17,18,19], LTα [19,48], and LTβ [49] have been generated and extensively used to establish the function of the various members of the TNF family. The complex relationships between the TNF-family members and their respective receptors were previously reviewed in detail [50].

Furthermore, the TNF signaling pathways were previously analyzed in detail and summarized [51]. While it is not part of this review, it should be pointed out that TNF signaling is linked to various pathways such as NfκB, the mitogen-activated protein (MAP) kinase, and the extracellular-signal-regulated kinase pathways that can run parallel or even compete. This explains why the functions of TNF are strictly context-dependent. The identification of a molecular domain near the C-terminus of the TNFR-1 receptor that induced cytotoxicity and was weakly homologous to a domain of the apoptosis-inducing Fas receptor was a groundbreaking development [52]. This domain was consequently termed a death domain and shown to define one major aspect of the extrinsic pathway of apoptosis. In the context of a potential TNF function in cancer immunity, this discovery offered a first insight into the possible mechanisms that allowed this cytokine to go beyond inflammation and have a role in tumor immunity.

However, even without a specifically defined mechanism, the assumption that TNF could act as a cytotoxic molecule was central to the hypothesis that this cytokine was effective as an anti-tumor agent. In fact, its cytotoxic role had been investigated for many years before the death domain of TNFR-1 was identified. In this context, early experiments identified TNF as a molecule expressed by activated lymphocyte blasts that was cytotoxic to L cells, transforming mouse fibroblasts [23]. TNF was also described as one of a variety of cytotoxic molecules produced by activated macrophages [13]. The cytotoxic activity of TNF was mostly shown to be dependent on or enhanced by the treatment of target cells with metabolic toxins. For example, the concentration and time-dependent cytolysis of L-929 cells by rabbit TNF was significantly increased by actinomycin D or cycloheximide treatment [53]. Similar observations of an increase in sensitivity were made in a second transformed cell line, WEHI 164 cells, after treatment with actinomycin C [54]. Additionally, it was shown that even primary fibroblasts responded in a similar fashion with an increased sensitivity to cytolysis by TNF when treated by act C or cycloheximide [55]. These experimental findings were used as the basis for assays that allowed the detection of the presence of TNF in serums or conditioned cell supernatants [54,56] and ultimately made purification and cloning possible.

## 3. TNF in Cancer Treatment

While previous experiments showed that TNF by itself was a weak cytotoxic agent with limited cytotoxic activity and a central involvement of TNF in the inflammatory response became clear very early in the investigation [1,11,57,58], which led to the development of powerful anti-inflammatory drugs [22], the interest in this cytokine as a cancer drug unabatedly continued. The cloning and expression of murine and human TNF made well-defined in vitro and in vivo experiments possible, and the question if TNF was biologically relevant in the immune response to tumor biology could finally be addressed.

Initial in vivo experiments were carried out in experimental settings such as in rodent models using murine or human TNF, and indeed, it was shown that high doses of TNF resulted in the selective necrosis of tumors. However, it also became clear that while mouse experiments that used high doses of human TNF were somewhat promising, using murine TNF in this context caused side effects that were comparable to inoculation with endotoxins, seemingly contradicting the earlier experience with Coley’s toxin [59] and thus preventing a quick transfer of TNF treatment to humans.

This problem to directly use TNF was underlined in early clinical trials using the cytokine as a monotherapy for various, mostly advanced cancers, which revealed very strong, dose-dependent side effects of TNF resembling endotoxic shock syndrome, such as fever, nausea, hypotension, and others [60,61,62,63]. TNF as a single agent was used in 18 experimental phase I treatments and ten phase II treatments. Their targets, the doses used, and the side effects of the treatment had been comprehensively summarized in detail up to 2011 [64]. Furthermore, TNF was used in combination with other cytokines, e.g., TNF and IFN γ [65], and TNF and IL-2 [66,67]. Eighteen of these phase I trials were also summarized [64]. Neither TNF alone nor in combination could claim to be a success in the published experimental trails and did not show anti-tumor effects that were significant or would go beyond anecdotal responses [64]. Since there is now a better understanding of the complex tumor microenvironment (TME) of solid tumors, it may be worthwhile to revisit the use of TNF to improve treatment efficacy [68].

While the use of TNF in therapy has been hampered by side effects and a lack of anti-tumor activity, attempts have been made to both reduce toxicity by means of the molecular modification of the cytokine and to increase tumoricidal activity by way of directly delivering TNF to the tissues. Decreasing toxicity was achieved by combining TNF with polyethylene glycol, which reduced the detrimental effects of the conjugated protein in vivo in a recurring tumor model in dogs [69]. While off-target effects remained to some extent, the toxicity was reduced, which allowed for a higher dose and better anti-tumor efficacy [70].

A very different approach to modulate the side effects of TNF, while still efficiently fighting soft tissue sarcomas and melanomas, has been developed that targets the delivery of the cytokine using isolated limb perfusion (ILP). After the vasculature of an extremity was isolated to avoid unwanted systemic effects, peripheral tissues were directly inoculated with high doses of TNF, mostly in combination with melphalan and sometimes IFN-γ [71,72,73,74,75]. This strategy has been used preoperatively to address the tumor size before surgery and has improved the chances to salvage the limb, even in the case of advanced and normally unresectable tumors [76]. In a systematic review, Moreno-Ramirez et al. examined the treatment of unresectable melanomas as defined by histological means and analyzed 22 studies with a total of 2018 ILPs [77]. In most cases, TNF was used in combination with melphalan. The treatment resulted in a 5-year survival time of 36%. No comparison with other methods was possible because by definition patients undergoing ILP have an unresectable disease. An accumulation of the chemotherapy drug melphalan in the tumor tissue [78] and the increased destruction of tumor vasculatures due to the presence of TNF were identified as underlying mechanisms [79].

In summary, while mouse experiments still supported the tumor-modifying action of TNF, the notion of a direct strong cytotoxic function had to be abandoned. Furthermore, a transfer to the human condition was not straightforward due to significant off-target effects and could only be used as a predominantly palliative measure to treat unresectable tumors using ILP.

## 4. Potential Mechanisms

Beyond primary cytotoxicity, a plethora of indirect actions of TNF must be considered as potential mechanisms that could modulate tumor growth, since many TNF functions are strongly context-dependent. In fact, after it was shown that TNF is predominantly produced by macrophages after activation [7,13], it became clear from the current perspective that Coley’s classical toxin did not work via inducing TNF cytotoxicity but via indirect means [80]. Two indirect TNF-dependent anti-tumor activities will be highlighted below.

### 4.1. Angiogenesis and Hemorrhagic Necrosis

One of the earliest activities of TNF that was elucidated was the activation of vascular endothelial cells [81]. This research led to the discovery of a strong induction of the angiogenic cytokine vascular permeability factor/vascular endothelial growth factor by TNF, and consequently, a pro-tumor function of TNF in tumor angiogenesis and vascular permeability [82,83]. In a separate line of research, the role of TNF in the hemorrhagic necrosis of tumors due to deterioration of their blood supply was recognized. Earlier work had shown that an injection of endotoxins into Bacillus Calmette–Guerin (BCG)-infected mice caused hemorrhagic necrosis, as we know now due to the release of TNF by macrophages [7]. This was confirmed in a mouse model of Meth A sarcomas that showed strong hemorrhagic necrosis after an intralesional injection with TNF [25]. The underlying mechanisms were shown to be congestion of the vasculature at 4–6 h with thrombus formations and blood circulation blockage at 24 h. In this study, the effect was attributed to direct anti-vascular cytotoxicity, despite in vitro results to the contrary [84].

The systemic treatment of mice with SA1 sarcoma with high doses of TNF was reliable at causing a tumor-hemorrhagic reaction that resulted in the destruction of greater than 75% of the tumor’s center in 24 h with the development of numerous hemorrhages in the tumor’s vascular bed [85].

### 4.2. TNF Is Linked to the Presence of M2 Macrophages in Tumors

Counterintuitively, in models of angiogenesis in rat corneas and developing chick chorioallantoic membranes, low doses of TNF induced blood vessel formation. It was conclusively demonstrated that activated macrophages were the source of this angiogenic effect [86]. This activity was also detectable in the vascularization of tumors. TNF was directly expressed in tumors by tumor cells [87,88] but predominantly by tumor-associated macrophages (TAMs). These types of macrophages were shown to be the dominant immune cell type in solid tumors [89,90,91,92]. These TAMs exhibited an alternative activation that resulted in a significantly changed activation pattern with other signature molecules, such as arginase being expressed [93]. This macrophage ‘‘polarization’’ in cancer resulted in a balance between phenotypes: the inflammatory M1-type versus the alternatively activated M2-type with a clear link of M2 macrophages to pro-tumor activities such as vascularization. Interestingly, in the context of this review, in animal models, TNF was essential for suppressing the number of M2 tumor macrophages and improved the chances of a positive outcome [94].

This surprising influence of TNF on macrophage polarization was confirmed in infection models [4].

## 5. Development of Anti-TNF Drugs

The strong proinflammatory activity of TNF was recognized early [57], as was the possibility to block this response using an anti-TNF serum [95]. The strong anti-inflammatory action of anti-TNF antibodies [96] and TNF-R fusion proteins [37] was investigated in more detail in various experimental models, such as in experimental autoimmune encephalomyelitis [97,98] and models of parasitic [99] and bacterial infections [100]. These observations encouraged various groups to develop and test therapeutics based on either anti-TNF antibodies or a combination of TNF-R and the Fc part of IgG. The three first anti-TNF biologicals that have been in clinical use for more than two decades will be discussed in more detail (Table 1).

### 5.1. Infliximab

The first anti-TNF biological available for clinical use was infliximab (trade name Remicade), which was based on the chimeric mouse–human antibody cA2 [101]. This chimeric antibody cA2 combined a murine variable region with specificity for human TNF with a human constant region and was developed by Centocor (US) (later Janssen-Biotec) in collaboration with the Department of Microbiology and the Kaplan Cancer Center, New York University Medical Center, New York.

Extensive investigations into the binding signature of cA2 and the blocked activities showed that it bound and neutralized soluble and membrane TNF but not lymphotoxins and inhibited cachexia and TNF-induced mortality. It also bound Fc receptors and could induce antibody-dependent cellular and complement-dependent cytotoxicity [102,104]. Concurrently, with these studies, initial clinical trials in RA patients showed a strong anti-inflammatory action [105] that could be maintained by repeated administration of the antibody without causing adverse effects [106]. Interestingly, it was demonstrated that the rapid onset of the improvement of the disease state of RA patients was due to an effect of infliximab on nociceptive brain activity, while inflammatory markers were not affected at this early point in therapy, demonstrating the wide and unexpected range of effects of blocking this cytokine [107].

### 5.2. Etanercept

At the same time that infliximab was developed, a TNF-R-2 immunoglobulin Fc-domain fusion protein was generated by Bruce Beutler’s laboratory at the Howard Hughes Medical Institute, University of Texas Southwestern Medical Center, Dallas [108]. The fusion protein was named etanercept and the rights were sold to Immunex. The drug is now marketed under the trade name Enbrel by Amgen. Since its binding is based on TNFR-2 in combination with the Ig–Fc domain, it additionally binds LTα in contrast to antibody-based drugs [109].

Early trials of etanercept in patients with RA showed that the drug had a strong anti-inflammatory action, especially in combination with methotrexate [110]. The fusion protein was approved by the FAA for the treatment of RA in 1998.

### 5.3. Adalimumab

The anti-TNF blocker adalimumab (trade name Humira) was developed in a collaboration between BASF Bioresearch Corporation and Cambridge Antibody Technology using phage-display technology to derive human antibodies for clinical development and therapy from gene libraries [103]. Adalimumab has been shown to have an activity spectrum comparable to infliximab and does not bind to LTα. It was tested in multiple clinical trials as a treatment for RA and was found to be highly effective with or without concomitant methotrexate [111,112]. The antibody was FAA-approved for RA therapy in 2002.

## 6. Target Pathologies

The affinities, avidities, and Fc-dependent complement activation of infliximab, etanercept, and adalimumab were quantified and compared in a comprehensive study [113]. While the affinity of the three inhibitors to soluble TNF was comparable, the avidity of etanercept was 10- to 20-fold higher. In contrast, the binding of membrane TNF was again comparable. Interestingly, in the context of this study, complement-induced cytotoxicity could be induced in transfected target cells but not in activated human PBMCs.

Anti-TNF drugs have been meanwhile approved for the treatment of various chronic inflammatory conditions of autoimmune origin, such as RA, polyarticular juvenile idiopathic arthritis, psoriatic arthritis, ankylosing spondylitis, and plaque psoriasis. In contrast to the TNFR-2-based anti-TNF fusion protein etanercept, the antibody-based drugs infliximab and adalimumab have been additionally approved for the treatment of Crohn’s disease.

## 7. TNF Inhibition Increases the Chance of Some Infectious Diseases

The high efficacy of these drugs against inflammatory conditions caused a fast increase of usage, but it became clear that there were unexpected adverse side effects. It was shown early in the investigation of the various roles of TNF that the maintenance of the protective granulomas in *M. tuberculosis* infection was dependent on the interaction of TNF with TNFR-1 [100]. This was confirmed in a mouse tuberculosis infection model in a TNF-deficient mouse strain [114]. Four years after the approval of infliximab and etanercept, more than 300 cases of tuberculosis were linked with the infliximab treatment of RA [115]. In 2004, a large collection of data showed that treatment with infliximab resulted in a higher risk of tuberculosis reactivation than the use of etanercept. While tuberculosis was the most frequently reported infection, other granulomatous infections, such as histoplasmosis or listeriosis, were also elevated by a total of 3.25-fold. This was explained by an as-yet unexplained difference of the mechanism of action of the two inhibitors [116]. Further attempts to pinpoint these differences in a mouse tuberculosis model showed that while earlier work had shown a dramatically superior neutralization efficiency of the TNFR fusion protein as compared with the anti-TNF antibody [117], treatment of chronically and acutely infected mice with the antibody inhibitor resulted in the death of both groups. After the TNFR-2 fusion protein treatment, only acutely infected mice died. An analysis of the granulomas showed that in chronic infection, the anti-TNF antibody penetrated the tissue deeper and was retained for an extended period of time as compared with the TNFR-2 fusion protein [118].

A comprehensive concluding study that involved advanced computer modeling yielded three important mechanistical observations that could explain the differences. First, a TNF inhibitor (TNFi) that binds to membrane TNF, such as infliximab, significantly reduces granuloma formation. Second, the antibody-based inhibitor shows strong differences in binding kinetics to TNF and tissue permeability causing reactivation, while apoptotic, cytolytic, and pharmacokinetic changes are not essential. Third, host-specific factors such as TNF-induced NF-κB activation that maintains granuloma function is still supported during etanercept but not infliximab treatment [119]. Finally, not discussed in this study but in other independent studies, etanercept also blocks LTα which can modulate the immune response to *M. tuberculosis* while signaling through the LTβR [120].

## 8. Is There a Role of TNF Inhibition in the Increase of Malignancies?

While the increase in human granulomatous diseases after TNF inhibition was identified and analyzed in mouse models with relative ease, the answer to the question if the neutralization of TNF increases the chance to receive a cancer diagnosis was originally based on anecdotal evidence and the historically perceived role of TNF. Despite a large number of investigations, the evidence is still controversially discussed. While infliximab and adalimumab are based on anti-TNF antibodies and bind only to TNF, etanercept is based on a TNFR-2-Ig fusion protein that can bind TNF and LTα [109], which could have consequences, since LTα potentially has a TNF-independent role in tumor control [121].

Because TNFi was first approved for the treatment of RA, the majority of initial investigations encompassed only small numbers of patients with short treatment phases. The first analysis of these data pointed to an increase in the incidence of lymphomas [122]. These preliminary studies were taken to the next level once more data about large patient numbers with a sufficient treatment duration were available. A study of 18,572 patients with RA who had been enrolled in the National Data Bank for Rheumatic Diseases (NDB) did not show a significant increase in lymphoma cases [123]. This study was updated in 2007 after additional data had been obtained and analyzed. The outcome of no increased incidences of lymphoma remained unchanged [124].

Similarly, two Swedish-population-based studies that were linked with the Swedish Nationwide Cancer and Census Registers, and differentiated between solid cancers [125] and hematopoietic malignancies [126] in three RA cohorts (one with prevalent pathology, admitted to hospital between 1990–2003 (n = 53,067); one incident cohort, with a cancer diagnosis between 1995–2003 (n = 3703); and one RA cohort treated with TNF antagonists between 1999–2003 (n = 4160)), was followed up for cancer occurrence through 2003. Neither for solid tumors, nor lymphomas, nor leukemia could a significantly elevated incidence be demonstrated. The extensive dataset in the Swedish cancer registers was used again in 2018 with a larger patient cohort to analyze TNF biologicals in cancer recurrences [127]. Of 467 RA patients who started treatment with TNFi with an average of 8 years after a cancer diagnosis, 42 had recurrences, while out of 2164 patients with a matched history, 155 had recurrences. These outcomes suggested that treatment with TNFi was not in any significant way responsible for cancer recurrence [127].

A more recent study analyzed the consequences of the treatment of 18,000 RA patients with TNFi and rituximab (RTX), and specifically the influence on 425 previous malignancies [128]. To compare recurrences, the authors used events per 1000 person-years and found that the TNFi cohort had 33.3 events/1000 person-years, while in the RTX cohort, there were 24.7 events/1000 person-years. A caveat of this investigation is the low number of cases, but within these limitations, past cancer did not predict future malignancies within these treatment regimens in a statistically significant way [128].

Finally, a study published in 2021 showed a relatively weak correlation with non-melanoma skin cancer and non-Hodgkin lymphoma and recommended in increased frequency of skin checks in older patients that used TNFi treatment regimens [129].

In the first meta-analysis, 144 potentially interesting publications that used TNFi to treat RA were analyzed. Because of various reasons, such as the use of TNFI based on TNFR-2 or the lack of a randomized control trial design, the number was reduced to 15. A duration of the treatment that was considered too short excluded another six trials. Nine clinical trials encompassing 5014 patients who had been randomized into treatment or control groups were analyzed [130]. This study that pooled earlier data showed a low, albeit significant increase in the risk of malignancies. A later systemic review clearly demonstrated that the use of TNFi was not correlated with an increase in the risk of overall malignant cancer, lymphoma, or melanoma [131] after the intrinsically increased risk of malignancies in autoimmune diseases was considered. This baseline bias towards malignancies in autoimmune diseases has been repeatedly shown in RA [132,133,134,135] and had to be taken into account before coming to final conclusions. Concurrently, with these observations of the adverse effects of TNF antagonists, TNFi, in this case Enbrel, was used as an experimental drug to interfere with progressive metastatic breast cancer [136] and ovarian cancer (inflammation) [137]. Both studies showed a low-level activity that could not be conclusively confirmed.

## 9. Conclusions

The research journey into the role of TNF in immunology in general and cancer immunology in particular started with huge expectations [11]. This went so far that the cloning of human TNF, which had been achieved and published in 1984 [25], was examined in the general press around the globe, and its therapeutic potential in cancer treatment was hailed as the coming of a new era. Very quickly, the molecule was advanced to preliminary clinical studies [138], which disappointed [139]. This could have been the end of it but for the discovery of the dominant proinflammatory aspect of TNF and the therapeutic effects of blocking it. This made TNF a major target in chronic inflammatory diseases, and its neutralization using biologicals such as antibodies in a clinical setting provided the first insights into the use of this new group of drugs. Thirty-eight years after the successful cloning of TNF, we can now conclusively answer the question if TNF has a role in the induction of or the immunity to malignancies. The fact that the widespread blocking of such a central proinflammatory cytokine in a plethora of inflammatory diseases did not significantly initiate malignancies underlines the fact that TNF is not directly involved in tumorigenesis or tumor recurrence and is unlikely to have a direct role in tumor immunity. However, this insight has not ended the interest in TNF. New research using nanoparticles for drug delivery into solid tumors has shown that TNF can act as a facilitator of a more efficient drug delivery into TME [68] with breast cancer as a specific example [140]. A recent review summarized the various nanoparticles in use and their specific targets [141].

## Figures and Tables

**Table 1 cancers-14-05270-t001:** “Classical” anti-TNF biologicals.

Name	Brandname	Molecular Structure	FAA-Registered	
Infliximab	Remicade	Chimeric (human/mouse) IgG1 k anti-TNF mAb	1998	[101]
Etanercept	Enbrel	Recombinant fusion protein: human TNFR-2:IgG1-Fc	1998	[102]
Adalimumab	Humira	Human IgG1k anti-TNF mAb	2002	[103]

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
