# Peer review of "Tumor Necrosis Factor: What Is in a Name?"

_cancers, 2022, doi:10.3390/cancers14215270_

Round 1

Reviewer 1 Report

Thank you for letting me review your work. I hope the comments find you well and will aid in making the article the best it can be. 

Section 1-2: Readers might be familiar with TNFalpha and TNFbeta. Introduce the nomenclature in the text and state your focus and how you will refer to them in the text.

Line 18: introduce the short form TNF, Tumour necrosis factor (TNF)

Line 28: is meth-A a accepted naming/abbreviation or should it be Methamphetamine (meth-A)?

Line 54, 224, 227-228 etc.: first time mentioning the abbreviation? Spell it out, rheumatoid arthritis (RA), then use the abbreviation later.

Line 65: be consistent tumours or tumors

Line 74, 80, 81 etc.: why separate brackets around references?

Line 75, 225: TNFR, first time spell it out, TNF receptor. Be consistent in how to write TNFR or TNF-R

Line 96: TNFR1, the abbreviation has not been introduced. Make it clearer to the reader by introducing the names TNFR1 and TNFR2 while writing about p55 and p75.

Line 111-115: can you rewrite this in a shorter sentence or divide in several sentences?

Line 115: unabatedly, do you mean undoubtedly?

Line 125: in x2.

Table 2: have TNFR2 been introduced? se comment line 96.

Line 131: IFN g, can you use the correct gamma sign?

Line 137: wrong phrasing. Attempts have been made.

Line 239: M. Tuberculosis in italic on line 264.

Line 268: the abbreviation TB has not been introduced earlier.

Line 269: APA state that brand names but not generic names should be capitalized. Check the whole text.

326: the abbreviation TNFi has not been introduced earlier.

As stated in the abstract you provide a review for  “its name-giving role in cancer biology”, eg. the antitumor effects. But TNF has also been implicated as a tumor driver via chronic inflammation, autoimmunity, immune escape and angiogenesis. You mention it briefly in the mechanism section. Perhaps could you do a small recap with more focus on the pro-tumor effects in a section close to the end?

There is still interest in TNF and its role in cancer. Could you comment on the current focus of research about TNF and cancer? 

Author Response

Please find the response attached below.

Reviewer 2 Report

The review entitled “Tumour Necrosis Factor: What is in a Name?” is interesting and well written.

My remarks concerning the manuscript:

Line 18, 194,195, 206, 258, : correct capital letters

Line 29: what do you mean by “fast method of detection”?

Line 38: what do you mean by „journey”?

Line 54: “RA” - rheumatoid arthritis?

Line 125: double “in”

Line 128: in what period of time? Up to today? Ref 65 is from 2011, thus this knowledge should be updated.

What is missing in these first paragraphs is the explanation, how, from molecular perspective, the same molecule may act as a cytotoxic agent (anti-tumor) and in parallel activate immune cells.

Line 197-199: this sentence is not clear

Line 295: “increase”? in incidence? This sentence is too long and very confusing

Author Response

(The authors gave the same response as above.)
